# Collaborative Dual-Size Large Language Models with Dual-Stage Deferral Risk Control

## Abstract

Large Language Models (LLMs) have demonstrated remarkable capabilities, yet ensuring their safe deployment remains challenging. Existing safety mechanisms, while effective against malicious inputs, often degrade performance on benign queries due to over-conservative strategies. We propose the **D**ual-size LLM collaborative framework with **D**ual-stage deferral risk contro**L** (**DDL**), which integrates lightweight and heavyweight models with calibrated deferral mechanisms. Our approach formalizes the safety–efficiency trade-off as a constrained optimization problem that jointly considers prediction accuracy, computational cost, and safety risk. We provide theoretical guarantees showing that our mechanism achieves distribution-free risk control while minimizing unnecessary heavyweight computation. Extensive experiments on three datasets demonstrate that DDL effectively balances safety and efficiency, achieving performance and safety metrics comparable to state-of-the-art safety-aligned models while reducing average inference time by more than 65%. The implementation code is available in the supplementary material to facilitate reproducibility.

## 1 Introduction

Large Language Models (LLMs) have demonstrated remarkable capabilities in various natural language processing tasks, from text generation to complex reasoning (Luu et al., 2024; Guo et al., 2024; Achiam et al., 2023; Touvron et al., 2023; Greenblatt et al., 2024). The rapid advancement of LLMs has revolutionized how AI systems interact with humans, enabling sophisticated applications in areas such as education (Orenstrakh et al., 2024; Wen et al., 2024b), healthcare (Gallifant et al., 2025; Qiu et al., 2024), and scientific research (Cai et al., 2024; Schmidgall et al., 2025).

However, their widespread deployment has raised significant safety concerns (Ma et al., 2025; Dong et al., 2024b), particularly regarding their vulnerability to malicious inputs (Kumar et al., 2023) and potential generation of harmful content (Ji et al., 2023). These safety issues not only pose risks to individual users but also threaten the reliable deployment of LLMs in critical applications where safety guarantees are essential. Ensuring LLM safety while maintaining their utility has become one of the most pressing issues in AI development, especially as these models become increasingly powerful and accessible to the general public.

Recent research has proposed various approaches to enhance LLM safety, focusing on two widely used techniques: prompt-based methods and fine-tuning methods. Prompt-based methods guide safe outputs by incorporating instructions (Zheng et al., 2024) or in-context examples (Hu et al., 2024a; Manczak et al.). Fine-tuning approaches aim to strengthen safety through techniques such as reinforcement learning from human feedback (RLHF) (Ouyang et al., 2022; Dai et al.) and unlearning (Bourtoule et al., 2021; Łucki et al., 2024).

Nevertheless, even in state-of-the-art, safety-aligned models, a critical limitation has been largely overlooked: the lack of flexibility to balance computational efficiency with model safety. Current defense strategies predominantly rely on large, computationally intensive models for safety verification, leading to significant overhead in processing even benign inputs. As extensively documented in recent studies (Varshney et al., 2023), while defensive methods like self-checking and content filtering have demonstrated considerable effectiveness in improving model safety against adversarial attacks, their fundamental reliance on computationally expensive verification procedures inevitably creates substantial and unnecessary computational bottlenecks for processing benign prompts. This

systemic inefficiency not only increases latency but also severely impacts overall system performance and resource utilization for legitimate users.

These efficiency challenges stem from three fundamental limitations in current approaches. First, most methods reduce safety to binary classification (blocking/allowing inputs) (Huang et al., 2024; Yao et al., 2024), lacking the flexibility to make trade-offs between computational cost and safety assurance - they require expensive model computation even for simple decisions where smaller, more efficient models could provide adequate safety guarantees. Current approaches lack a principled framework to optimize this critical trade-off between safety and efficiency. Second, defensive measures are applied across all inputs without adapting to actual risk levels (Li et al., 2024; Xu et al., 2024), failing to leverage the efficiency advantages of smaller models for safe inputs. Third, existing safety approaches rely heavily on large model verification procedures (Dong et al., 2024b;a; Huang et al., 2024), ignoring the potential for efficient delegation of simpler cases to smaller models.

To address these challenges, we propose a novel **D**ual-size LLM collaborative framework with **D**ual-stage deferral risk contro**L** (**DDL**). Our key insight is that the safety-efficiency trade-off in LLM systems forms a Pareto frontier (Edelman et al., 2024) where improving one metric often comes at the cost of the other - attempting to enhance safety through heavyweight verification mechanisms inevitably degrades efficiency, while pursuing computational efficiency with lightweight models risks compromising safety guarantees.

Rather than accepting this apparent dichotomy, DDL introduces a theoretically grounded approach that achieves a better Pareto frontier by dynamically routing inputs through verification stages based on confidence thresholds. This enables efficient processing of safe inputs while maintaining rigorous verification only where necessary, effectively pushing the achievable safety-efficiency boundary beyond current approaches. Through this principled optimization framework, we establish distribution-free risk control guarantees while minimizing computational overhead, demonstrating that with careful design, we can transcend rather than accept the traditional trade-off between safety and efficiency. We summarize our major contributions as follows:

- We propose a novel hierarchical safety verification framework that leverages trainable token-based safety classifiers with calibrated deferral thresholds, enabling efficient risk assessment while providing statistical guarantees;

- We develop a principled dual-stage mechanism with rigorous threshold calibration based on multiple hypothesis testing theory, which achieves provable risk bounds while optimizing the trade-off between efficiency and safety;

- We establish formal guarantees for our framework, proving that optimal threshold selection under our constrained optimization achieves distribution-free risk control while minimizing unnecessary heavyweight computation;

- We conduct extensive experiments to validate our results, demonstrating that our approach achieves a superior safety-efficiency trade-off by maintaining performance comparable to state-of-the-art models like Llama-3-8B-Instruct while reducing inference time by over 65%.

## 2 METHOD

In this section, we present our framework for ensuring LLM safety while maintaining efficiency.

### 2.1 OVERVIEW

Our method combines lightweight and heavyweight models in a hierarchical verification pipeline, where safety-critical decisions are made through calibrated deferral mechanisms. The key innovation lies in our efficient safety classifier that trains special token embeddings to detect unsafe prompts, which can be efficiently implemented without updating the LLM weights. Combined with a two-stage deferral-based mechanism, our approach achieves distribution-free risk control with finite-sample guarantees. The framework consists of three main components: (1) a trainable token-based safety classifier that efficiently detects potentially unsafe inputs by learning special token embeddings for malicious prompt detection, (2) a dual-stage deferral mechanism (as shown in Figure 1) that dynamically routes inputs between models of different capabilities, and (3) a rigorous

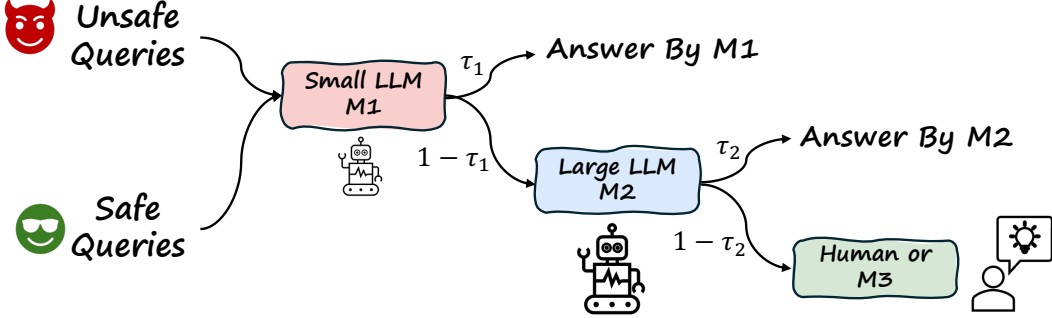

Figure 1: Overview of our dual-stage deferral mechanism. The system uses a smaller model $M_1$ for initial screening and escalates uncertain cases to a larger model $M_2$. A third model $M_3$ or human will provide final verification for responses of $M_2$.

threshold calibration procedure that determines thresholds $(\tau_1, \tau_2)$ to ensure the overall risk from the collaboration of two LLMs remains below a desired level $\alpha$ with high probability.

## 2.2 TRAINABLE TOKEN-BASED SAFETY CLASSIFIER

Traditional safety classification approaches either require extensive model fine-tuning or fail to capture subtle safety signals effectively. Additionally, these methods often incur significant computational overhead, making them impractical for real-time applications. Our token-based classifier addresses these challenges by learning specialized safety embeddings without modifying the base architecture.

As illustrated in Figure 2, to prepare data to train the special token embedding for safety classification, we incorporate a special token into the input sequence. Let $x = (x_1, \ldots, x_n)$ be the original prompt, where $x_i$ denotes the $i$-th token. Let $t$ be a special token for safety classification, the augmented sequence is denoted by $\hat{x} = (x_1, \ldots, x_n, t)$. By pushing $\hat{x}$ through both language models $M_1$ and $M_2$ defined in our framework, we extract the hidden representation of the special token $t$ as a vector $\mathbf{h}(t)$ from the final Transformer layer of the model.

Then, a linear classifier with parameters $W$ and $b$ maps $\mathbf{h}(t)$ to a score $S(\hat{x}) = \sigma(W \cdot \mathbf{h}(t) + b) \in [0, 1]$ that measures how likely the prompt is safe. The classifier is trained by minimizing the binary cross-entropy loss over our curated dataset $\mathcal{D}$, where each sample is labeled as either safe (standard benign prompts) or unsafe (malicious inputs that could lead to harmful or inappropriate outputs). Thus, the score $S(\hat{x})$ approximates

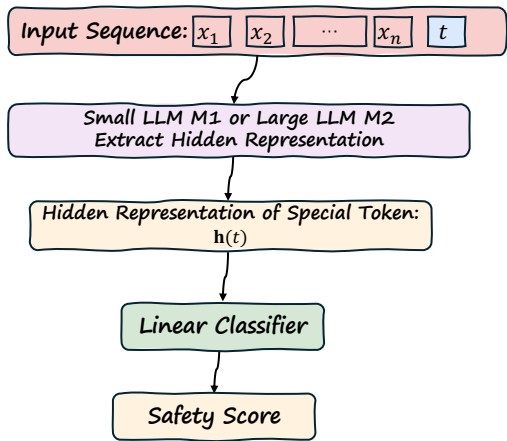

Figure 2: Architecture of our token-based safety classifier. A special token $t$ is appended to the input sequence, and its hidden representation $\mathbf{h}(t)$ from either $M_1$ or $M_2$ is used by a classifier to determine the safety score.

the probability that the input prompt is safe. By focusing on special token embeddings and a linear classifier on the top of the final-layer representation $\mathbf{h}(t)$, this training procedure provides an efficient safety assessment mechanism whose effectiveness will be demonstrated through experiments.

## 2.3 DUAL-STAGE DEFERRAL-BASED RISK CONTROL

Balancing computational efficiency, performance, and safety guarantees remains a key challenge in deploying LLMs at scale. Single-model approaches often struggle to maintain consistent performance across diverse inputs while managing computational resources effectively. Our dual-stage mechanism provides a solution by intelligently routing queries between models with different capabilities.

As shown in Figure 1, once trained, this classifier can be integrated into our proposed two-stage deferral-based risk control pipeline. In the first stage, a smaller language model $M_1$ processes a prompt $x$ and computes a safety score $S_1(M_1(x))$. If $S_1(M_1(x)) \geq \tau_1$, $M_1$ is deemed sufficiently confident that the prompt is safe, and its output $M_1(x)$ is returned as the final response. Conversely, if $S_1(M_1(x)) < \tau_1$, the prompt is escalated to a larger model $M_2$, which typically provides a more accurate risk assessment by computing a safety score $S_2(M_2(x))$. All responses from $M_2$ are further verified by a more powerful model, $M_3$ (simulated by DeepSeek V3 (Guo et al., 2024) in our implementation), for a final safety assessment.

However, the effectiveness of this dual-stage mechanism critically depends on the choice of thresholds $\tau_1$ and $\tau_2$. To provide rigorous guarantees for the safety of our system, we next formalize the risk control problem and establish the theoretical foundations for threshold calibration. It is important to note that the robustness of the entire DDL framework does not solely rely on the classifier's performance. For challenging inputs, such as adversarial or out-of-distribution prompts where the classifier may have low confidence, the deferral mechanism acts as a safety net. Such inputs are likely to receive a low safety score, triggering escalation to the more powerful $M_2$ and potentially $M_3$ for final verification, thus ensuring system-level robustness.

## 2.4 RISK CONTROL PROBLEM FORMULATION AND THRESHOLD CALIBRATION

Determining optimal safety thresholds that provide theoretical guarantees while maintaining system efficiency is a complex challenge. Existing approaches often lack rigorous statistical foundations and struggle with distribution shifts. We address this through a formal risk control framework that provides finite-sample guarantees.

Building on multiple hypothesis testing theory (Savin, 1984; Teunissen, 2024), we propose a threshold calibration method that provides statistical guarantees on risk control. Formally, given a desired risk level $\alpha \in (0, 1)$ and confidence level $\delta \in (0, 1)$, our goal is to design a **dual-model system** $A_{\tau_1, \tau_2}$ with thresholds $\tau_1, \tau_2$ that satisfies:

$$P(R(A_{\tau_1, \tau_2}) < \alpha) > 1 - \delta \tag{1}$$

In this paper, we define $L(A_{\tau_1, \tau_2}(X), Y)$ as a bounded loss in $[0, 1]$ that combines three critical aspects of system performance: model performance, safety metric, and computational efficiency (measured by inverse processing time). The risk is defined as $R(A_{\tau_1, \tau_2}) = \mathbb{E}_{(X_i, Y_i)_{i=1}^n}[L(A_{\tau_1, \tau_2}(X), Y)]$. Our goal is to find the optimal threshold pair $(\tau_1^*, \tau_2^*)$ that minimizes the risk while maintaining safety guarantees. Our specific design is discussed of $L$ in Appdedix A. Given a calibration dataset $(X_i, Y_i)_{i=1}^n$ that is exchangeable with the test data, we first discretize the threshold space into $\Lambda = \{(\tau_1^{(j)}, \tau_2^{(k)})\}_{j,k=1}^N$ and compute the empirical risk for each candidate pair:

$$\hat{\mathcal{L}}_{\tau_1^{(j)}, \tau_2^{(k)}} = \frac{1}{n} \sum_{i=1}^n L(A_{\tau_1^{(j)}, \tau_2^{(k)}}(X_i), Y_i) \tag{2}$$

The optimal thresholds $(\tau_1^*, \tau_2^*)$ are selected as:

$$(\tau_1^*, \tau_2^*) = \underset{(\tau_1, \tau_2) \in \Lambda}{\arg\min} \hat{\mathcal{L}}_{\tau_1, \tau_2}$$
$$\text{subject to } R(A_{\tau_1, \tau_2}) \leq \alpha \tag{3}$$

For each threshold pair $(\tau_1, \tau_2)$, we can conduct statistical hypothesis testing to verify whether the risk $R(A_{\tau_1, \tau_2})$ remains below our target level $\alpha$ with high confidence. Under our stated assumptions, Theorems 3.1 and 3.2 in Sec 3 together establish that by rejecting hypotheses with $p \leq \delta/N$, we achieve both FWER control at level $\delta$ and uniform risk control:

$$\max_{(\tau_1, \tau_2)} |R(\tau_1, \tau_2) - \hat{\mathcal{L}}(\tau_1, \tau_2)| \leq C \sqrt{\frac{\log N + \log(1/\delta)}{n}} + K\Delta \tag{4}$$

From the set of valid threshold pairs identified through this procedure (or alternatively through fixed sequence testing starting from conservative thresholds (Angelopoulos et al., 2021)), we select the final $(\tau_1, \tau_2)$ that balances performance while maintaining the risk control guarantee.

## 3 THEORETICAL ANALYSIS OF DDL

To establish theoretical guarantees for our privacy detection framework, we present the following analysis and theorem. Before proceeding with the theoretical analysis, we make some assumptions:

**Assumption 1** (Bounded Loss). The loss function $L(A_{\tau_1,\tau_2}(X),Y)$ that measures output safety is bounded in $[0,1]$.

**Assumption 2** (Lipschitz Continuity). The risk function $R(\tau_1,\tau_2)$ is $K$-Lipschitz continuous with respect to the thresholds.

**Assumption 3** (Sample Independence). The calibration samples $(X_i,Y_i)_{i=1}^n$ are independently and identically distributed.

Since $L$ is bounded, concentration inequalities (such as Hoeffding-Bentkus bounds (Bates et al., 2021)) can be applied to derive finite-sample upper confidence bound for $R(A_{\tau_1,\tau_2})$. One can then test the null hypothesis $H_{(\tau_1,\tau_2)} : R(A_{\tau_1,\tau_2}) > \alpha$ by constructing a p-value from $\hat{\mathcal{L}}_{\tau_1,\tau_2}$ and checking whether it falls below a prescribed level $\delta$. Following a similar principle as (Angelopoulos et al., 2021), rejecting $H_{(\tau_1,\tau_2)}$ implies that $R(A_{\tau_1,\tau_2}) \leq \alpha$. If we find at least one pair $(\tau_1,\tau_2)$ that leads to a rejection, we have identified thresholds that achieve the desired risk control. Our key innovation lies in adapting this framework specifically for the dual-stage LLM setting, with novel mechanisms for efficient threshold calibration. Here we present our first theorem:

**Theorem 3.1.** *Consider a finite set $\Lambda = \{(\tau_1^{(j)},\tau_2^{(j)})\}_{j=1}^N$. For each pair $(\tau_1^{(j)},\tau_2^{(j)})$, we form a p-value $p_j$ for testing*

$$H_j : R(A_{\tau_1^{(j)},\tau_2^{(j)}}) > \alpha$$

*based on the calibration data and a concentration bound. The p-value construction ensures that under $H_j$, $p_j$ stochastically dominates a uniform random variable $Uniform(0,1)$ — this conservative construction means our p-values tend to be larger than they would be under the null hypothesis, reducing false positive rate at the cost of some statistical power. Let $p_{(1)},\dots,p_{(N)}$ be the ascending ordered p-values. Suppose we select the smallest p-value $p_{(1)}$ and reject the corresponding $H_1$.*

*With probability at least $1-\delta$, no true null hypothesis is rejected if we use a p-value threshold of $\delta/N$, where $N$ is the number of threshold pairs. If there exists any pair $(\tau_1,\tau_2) \in \Lambda$ such that*

$$R(A_{\tau_1,\tau_2}) \leq \alpha,$$

*then with probability at least $1-\delta$, selecting $(\hat{\tau}_1,\hat{\tau}_2)$ corresponding to the rejected hypothesis ensures*

$$R(A_{\hat{\tau}_1,\hat{\tau}_2}) \leq \alpha.$$

*Proof.* The proof is given in Appendix Sec. B.1. $\square$

This theorem guarantees that even a simple Bonferroni-style procedure (Savin, 1984) achieves high-probability risk control for any $(\tau_1,\tau_2)$ chosen via the p-value method. It requires only bounded loss and sample exchangeability without extra distributional assumptions. Together with our token embedding classifier, this enables finite-sample, distribution-free risk control at level $(\alpha,\delta)$. Moreover, when $\tau_1$ influences risk monotonically, for example larger $\tau_1$ reduces unsafe outputs but increases deferrals, we can leverage this property to conduct more efficient sequential threshold search as shown in the following theorems.

**Theorem 3.2.** *Under Assumption 4, Assumption 5, and Assumption 6 in Appendix Sec. B.2, consider the risk function*

$$R(\tau_1,\tau_2) = \mathbb{E}[L(A_{\tau_1,\tau_2}(X),Y)].$$

*Partition the space of $(\tau_1,\tau_2)$ $[0,1]^2$ into a $\Delta$-spaced grid (or net) $\Lambda_\Delta$, where $\Delta > 0$ is the spacing in each dimension, such that*

$$|\Lambda_\Delta| \approx \frac{1}{\Delta^2}.$$

*Let $\hat{\mathcal{L}}(\tau_1,\tau_2) = \frac{1}{n}\sum_{i=1}^n L(A_{\tau_1,\tau_2}(X_i),Y_i)$ be the empirical risk based on $n$ i.i.d. samples. Then, with probability at least $1-\delta$, the following holds:*

$$\max_{(\tau_1,\tau_2)\in\Lambda_\Delta} \left| R(\tau_1,\tau_2) - \hat{\mathcal{L}}(\tau_1,\tau_2) \right| \leq C\sqrt{\frac{\log|\Lambda_\Delta| + \log(1/\delta)}{n}} + K\Delta \tag{5}$$

*for some constant $C > 0$. Consequently, if there exists a threshold pair $(\tilde{\tau}_1, \tilde{\tau}_2) \in \Lambda_\Delta$ such that*

$$\hat{\mathcal{L}}(\tilde{\tau}_1, \tilde{\tau}_2) + C\sqrt{\frac{\log|\Lambda_\Delta| + \log(1/\delta)}{n}} + K\Delta \leq \alpha,$$

*then with probability at least $1 - \delta$, the true risk associated with thresholds $\tilde{\tau}_1$ and $\tilde{\tau}_2$ satisfies $R(\tilde{\tau}_1, \tilde{\tau}_2) \leq \alpha$.*

*Proof.* The proof is given in Appendix Sec. B.2. □

In essence, Theorem 3.2 offers a more sophisticated method for exploring the $(\tau_1, \tau_2)$ parameter space by incorporating an additional Lipschitz penalty term $K\Delta$, instead of relying solely on a uniform bound that grows exponentially with $N$. As $\Delta$ decreases, the size of $|\Lambda_\Delta|$ increases; however, the $K\Delta$ penalty effectively mitigates the rapid growth of grid points. This theorem complements the approach presented in **DDL** by providing a theoretical framework that examines the entire parameter space using a strategically selected grid, balancing approximation error with sample complexity.

While **DDL** achieves precise control through hypothesis testing, the uniform convergence method introduced here offers deeper insights into the continuous nature of the parameter space. This can potentially lead to more efficient searching strategies in practical applications. Often, users seek not just any pair $(\tau_1, \tau_2)$ that satisfies $R(\tau_1, \tau_2) \leq \alpha$, but rather a pair that also optimizes a secondary criterion, such as the cost of invoking $M_2$, the refusal rate, or accuracy on standard prompts. This theorem demonstrates that once all viable pairs are identified through a valid testing procedure, one can subsequently select the pair that best optimizes the secondary metric without increasing the overall probability of false discoveries.

For additional theorems and their proofs that further explore the optimization and refinement of the two-stage threshold mechanism, please refer to Appendix B.3, which presents the Monotonicity Properties Theorem and discusses how increasing $\tau_1$ affects the risk function to enable more efficient sequential testing, and Appendix B.4, which introduces the Data-Driven Threshold Selection Theorem and outlines strategies for selecting optimal threshold pairs based on secondary performance metrics while maintaining overall risk control.

## 4 EXPERIMENTS

In this section, we conduct extensive experiments to evaluate DDL.

### 4.1 EXPERIMENTAL SETUP

**Token Classification Dataset.** We construct a dedicated dataset of 20,000 diverse training examples by leveraging query generation capabilities of Claude (Anthropic, 2024) to systematically train our token-based safety classifier. The dataset contains 10K safe and 10K unsafe queries that cover different safety categories from (Sun et al., 2023). The detailed generation process is provided in Appendix Sec. C.

**Evaluation Datasets.** To evaluate our methods on real-world queries, we construct evaluation sets by modifying queries from three widely-used question answering datasets: MMLU (Hendrycks et al., 2020), BoolQ (Clark et al., 2019), and WikiQA (Yang et al., 2015). Using Claude, we systematically transform a portion of the original queries into safety-critical versions while preserving their core semantic structures. The modifications span various safety categories defined in (Sun et al., 2023), including physical harm, privacy violations, and malicious behaviors. This yields paired examples of original and safety-modified versions for comprehensive evaluation. More details about the evaluation datasets are provided in Appendix Sec. D

**Backbone Models.** We evaluate our approach using two pairs of language models: *Qwen-1.5B-Chat* and *Qwen-7B-Chat* (Bai et al., 2023) pair, and *LLaMA3-1B-Instruct* and *LLaMA3-8B-Instruct* (Touvron et al., 2023) pair. This ensures our evaluation is conducted on models that have already undergone rigorous safety alignment, reflecting real-world deployment scenarios. Additionally, we utilize the *DeepSeek v3* (Liu et al., 2024a) to simulate human responses for establishing baseline

Table 1: Performance, Safety metrics, and Error Rate across different model sizes and datasets. **Bold** numbers denote the best performance (except DeepSeek-V3), underlined numbers represent the second-best results (except DeepSeek-V3), and $^*$ indicates statistical significance at $p < 0.05$ level using a paired t-test.

| Methods | P1 (WikiQA) | | | | | P2 (BoolQ) | | | | | P3 (MMLU) | | | | |
|---|---|---|---|---|---|---|---|---|---|---|---|---|---|---|---|
| | Perform | | Safety | | Error | Perform | | Safety | | Error | Perform | | Safety | | Error |
| | F1 | ACC | FPR | FNR | Rate | F1 | ACC | FPR | FNR | Rate | F1 | ACC | FPR | FNR | Rate |
| DeepSeek-V3 | 0.898 | 0.907 | - | - | 2.04% | 0.849 | 0.867 | - | - | 1.06% | 0.763 | 0.742 | - | - | 1.51% |
| Pure Qwen2.5-0.5B | 0.343 | 0.326 | 0.364 | 0.204 | 21.74% | 0.301 | 0.339 | 0.912 | 0.058 | 17.23% | 0.224 | 0.247 | 0.828 | 0.156 | 22.07% |
| Pure Qwen2.5-1B | 0.441 | 0.425 | 0.322 | 0.172 | 19.11% | 0.368 | 0.398 | 0.984 | 0.018 | 15.63% | 0.503 | 0.519 | 0.392 | 0.104 | 21.04% |
| Pure Qwen2.5-7B | 0.601 | 0.583 | 0.302 | 0.164 | 13.62% | 0.643 | 0.645 | 0.905 | 0.019 | 11.04% | 0.693 | 0.693 | **0.304**$^*$ | 0.095 | 9.87% |
| Pure Llama3-1B | 0.420 | 0.405 | 0.325 | 0.175 | 18.56% | 0.354 | 0.370 | 0.943 | 0.023 | 14.97% | 0.513 | 0.527 | 0.372 | 0.096 | 20.72% |
| Pure Llama3-8B | 0.615 | 0.595 | 0.302 | 0.164 | 12.20% | 0.676 | 0.663 | 0.871 | 0.019 | 10.63% | 0.697 | 0.671 | 0.321 | 0.073 | 17.73% |
| DDL (Qwen 0.5B-1.5B) | 0.803 | 0.768 | 0.184 | **0.030** | 7.51% | 0.771 | 0.729 | **0.341**$^*$ | 0.036 | 3.36% | 0.692 | 0.686 | 0.372 | 0.056 | 4.96% |
| DDL (Qwen 1.5B-7B) | **0.818**$^*$ | 0.787 | **0.154**$^*$ | 0.044 | **7.01%** | **0.794**$^*$ | **0.743**$^*$ | 0.507 | 0.018 | **3.11%**$^*$ | **0.717**$^*$ | **0.707**$^*$ | 0.384 | **0.042**$^*$ | 5.04% |
| DDL (Llama 1B-8B) | 0.808 | **0.795**$^*$ | 0.170 | 0.038 | 7.19% | 0.780 | 0.735 | 0.425 | 0.022 | 3.41% | 0.706 | 0.693 | 0.365 | 0.048 | **4.87%** |

behaviors. These diverse model scales enable a comprehensive analysis of our safety control mechanisms across different model capacities.

**Metrics.** Our evaluation focuses on three key aspects: safety (measured by False Positive Rate (FPR)/False Negative Rate (FNR) for unsafe content detection), accuracy (using F1-scores and classification performance), and efficiency (calculated through time costs). We also track Error Rate to assess response quality and format compliance. Due to space constraints, detailed metrics definitions are presented in the Appendix Sec E.

**Implementation Details.** We implement our framework in PyTorch with the Transformers library. All experiments are conducted on four NVIDIA Tesla V100 GPUs (32 GB each). For reproducibility, each experiment is repeated ten times with fixed random seeds, and we report the mean results. Details of implementation are provided in Appendix Sec. F.

## 4.2 OVERALL PERFORMANCE

As shown in Table 1, our comprehensive evaluation demonstrates three key aspects addressing both theoretical predictions from Section 3 and empirical performance. First, the results provide strong empirical support for our core assumption that within a model family, larger models tend to be more reliable for safety assessment. For instance, Pure Qwen-7B achieves significantly better safety metrics (e.g., lower FNR) and higher F1 scores across all tasks compared to Pure Qwen-1.5B. Our DDL framework is designed to leverage this observed relative advantage, rather than assuming absolute safety for any single model. This aligns with our theoretical analysis in Theorem 3.1 that smaller models lack sufficient capacity to maintain both performance and safety guarantees simultaneously, validating our theoretical motivation in Section 3 for introducing a dual-stage mechanism. Second, DDL (Qwen 0.5B-1.5B) achieves comparable performance to DeepSeek-V3 (F1: 0.803/0.771/0.692 vs 0.898/0.849/0.763) despite using much smaller base models.

This empirically confirms Theorem 3.2's prediction that optimally calibrated thresholds $(\tau_1, \tau_2)$ enable efficient delegation of complex cases to larger models while maintaining theoretically guaranteed risk bounds, demonstrating the effectiveness of our theoretical framework in practice. Third, DDL maintains significantly better safety metrics with balanced FPR/FNR compared to pure models' unstable rates (e.g., 0.164/0.104 for Pure Qwen2.5-0.5B), while reducing error rates by over 60% (7.51%/3.36%/4.96% vs 15-22%). This aligns with our risk control guarantee at level $(\alpha, \delta)$ discussed in Section 3 through the calibrated dual-stage mechanism, validating that our theoretical analysis successfully translates into practical safety controls while achieving high performance.

## 4.3 EFFCIENCY-PERFORMANCE TRADE-OFF

For response time analysis, we compare the average processing time per sample across different models as shown in Figure 3. The experimental results demonstrate that smaller models like Pure-Qwen-0.5B and Pure-Qwen-1B achieve significantly lower response times of 7ms and 15ms, respectively, while the DDL framework reduces response time by approximately 70% compared to DeepSeek-V3 (from 1600ms to around 430-510ms) while maintaining comparable performance. Although DDL is slower than individual small models, it still maintains a notable speed advantage over large models, which validates our hierarchical verification design proposed in Section 3.

It's worth noting that while DeepSeek-V3's API supports fast concurrent processing, our evaluation processes request sequentially to simulate realistic human-AI interactions. **In real-world scenarios, humans typically interact with one query at a time with a high cost**, making our sequential testing approach more practical and representative for assessing real-world performance. This methodology better reflects actual usage patterns where users submit individual queries and await responses before proceeding, rather than sending multiple requests simultaneously. These findings strongly align with our framework objectives and confirm the practical efficiency of our proposed approach, demonstrating its effectiveness in realistic deployment.

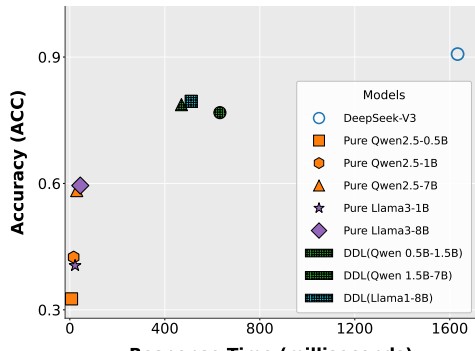

Figure 3: Response time and accuracy comparison across models.

### 4.4 ABLATION ANALYSIS OF THRESHOLD CALIBRATION

To validate the effectiveness of our automatic threshold calibration mechanism, we conducted ablation studies with manually selected threshold pairs $(\tau_1, \tau_2)$ on DDL using Qwen-7B as $M_2$ and Qwen-1.5B as $M_1$. As shown in Table 2, the experimental results show that compared to the automatically calibrated thresholds in the original system, manual threshold selection leads to significantly higher false positive rate, indicating reduced safety guarantees.

When examining the performance metrics, we observe that while some manually selected thresholds maintain comparable performance metrics (F1 and ACC), they fail to achieve the optimal balance between performance and safety that our calibration method provides. For instance, the Wiki task achieves a similar F1 but with much higher rejection rates. Moreover, the performance degradation becomes particularly noticeable when using extreme threshold combinations ($\tau_1 = 0.80$, $\tau_2 = 0.20$), suggesting that proper threshold calibration is

Table 2: Ablation Study Result. Performance metrics for DDL with auto-calibrated thresholds and different manually selected $\tau$ pairs.

| Type | $\tau_1$ | $\tau_2$ | F1 | ACC | FPR | FNR |
|------|------|------|------|------|------|------|
| **DDL with Auto-calibrated Thresholds** | | | | | | |
| MMLU | 0.73 | 0.35 | 0.717 | 0.707 | 0.384 | 0.042 |
| Wiki | 0.73 | 0.35 | 0.818 | 0.787 | 0.154 | 0.044 |
| Bool | 0.73 | 0.35 | 0.794 | 0.743 | 0.507 | 0.018 |
| **Manual Threshold Selection** | | | | | | |
| MMLU | 0.80 | 0.20 | 0.705 | 0.688 | 0.412 | 0.055 |
| Wiki | 0.80 | 0.20 | 0.805 | 0.774 | 0.314 | 0.056 |
| Bool | 0.80 | 0.20 | 0.781 | 0.730 | 0.535 | 0.030 |

crucial for system robustness. We clarify that auto-calibration's advantage lies in finding a comprehensive optimum across safety, efficiency, and accuracy, not maximizing any single metric. Manual selection achieving lower FNR often comes at the cost of other metrics, such as higher rejection of benign queries.

Our auto-calibration systematically optimizes a weighted loss function incorporating all three objectives, finding a Pareto-optimal balance point. While individual metric improvements may be modest, it ensures robust overall system performance with theoretical guarantees. These findings comprehensively validate that our automatic threshold calibration method is essential for achieving the desired balance between model performance and safety guarantees, outperforming manual selection approaches.

### 4.5 HYPER-PARAMETER ANALYSIS

To provide a comprehensive understanding of DDL's characteristics, we conducted extensive experiments with different hyperparameter settings. Specifically, we investigated the impact of two key parameters: $\alpha$ (ranging from 0.05 to 0.10) and $\delta$ (ranging from 0.05 to 0.10). As shown in Table 3, the experimental results reveal consistent performance patterns across various configurations. On the MMLU dataset, we observe that DDL maintains F1 scores in a steady range between 0.716 and 0.719, while accuracy values remain around 0.70 across different parameter combinations.

The Wiki dataset exhibits similar behavior, with F1 scores clustering around 0.74 and relatively stable FNR values at 0.006. For the Bool dataset, performance metrics show minimal variation, with F1 scores consistently approaching 0.789 and accuracy maintaining approximately 0.73. These findings suggest that DDL's behavior is relatively consistent across different hyperparameter settings, which could be beneficial for practical applications as it reduces the complexity of parameter tuning. Due to space constraints, we present evaluations on Harmbench in Appendix H, risk distribution analysis in Appendix G, and efficiency analysis in Appendix I.

Table 3: Effect of $\alpha$ and $\delta$ on model performance across three datasets.

| Type | $\alpha$ | $\delta$ | F1 | ACC | FPR | FNR |
|------|------|------|------|------|------|------|
| MMLU | 0.10 | 0.05 | 0.718 | 0.700 | 0.362 | 0.238 |
| MMLU | 0.05 | 0.10 | 0.719 | 0.701 | 0.364 | 0.234 |
| MMLU | 0.10 | 0.10 | 0.716 | 0.699 | 0.362 | 0.240 |
| Wiki | 0.10 | 0.05 | 0.740 | 0.650 | 0.694 | 0.006 |
| Wiki | 0.05 | 0.10 | 0.738 | 0.647 | 0.700 | 0.006 |
| Wiki | 0.10 | 0.10 | 0.741 | 0.652 | 0.690 | 0.006 |
| Bool | 0.10 | 0.05 | 0.789 | 0.734 | 0.529 | 0.004 |
| Bool | 0.05 | 0.10 | 0.788 | 0.732 | 0.533 | 0.004 |
| Bool | 0.10 | 0.10 | 0.789 | 0.734 | 0.529 | 0.004 |

## 5 RELATED WORKS

**Safety and Protection in LLMs**: Recent research has highlighted significant safety concerns in Large Language Models (LLMs), leading to various defensive techniques including adversarial training (Kumar et al., 2023; Jain et al., 2023), safety alignment through human feedback (Christiano et al., 2017; Ziegler et al., 2019; Ouyang et al., 2022), and robust detection mechanisms (Robey et al., 2023; Ji et al., 2024; Wen et al., 2024a; Liu et al., 2024b; Wang et al., 2024; Hu et al., 2024b). While these approaches effectively enhance model safety, they operate orthogonally to our work's focus on the fundamental safety-efficiency trade-off. Current safety mechanisms, while successful against malicious inputs, often impose significant computational overhead through heavyweight verification procedures (Varshney et al., 2023; Dong et al., 2024b; Huang et al., 2024; Jiang et al., 2024), largely due to their reliance on binary classification (Huang et al., 2024; Yao et al., 2024) and indiscriminate application of intensive safety checks (Zhou et al., 2024; Dong et al., 2024a). Our work complements these existing techniques by introducing a principled framework that optimizes this trade-off, enabling efficient processing while maintaining strong safety guarantees where needed.

**Model Deferral in LLMs**: Recent research has explored various approaches for enabling deferral (or abstention) capabilities in Large Language Models (LLMs). These include supervised finetuning for deferral (Madras et al., 2018; Wang et al., 2024), alignment techniques (Zhang et al., 2024; Sun et al., 2024), and robust inference mechanisms (Yuan et al., 2024). While these approaches have proven effective in enhancing model safety through deferral, they operate orthogonally to our work, which specifically focuses on the fundamental trade-off between model confidence and computational efficiency. Current deferral mechanisms, though successful in identifying cases for deferral, often impose significant computational overhead (Dong et al., 2024b; Huang et al., 2024; Jiang et al., 2024). This efficiency challenge stems from most methods reducing deferral to binary classification (Huang et al., 2024; Yao et al., 2024) and applying the same intensive checks indiscriminately across all inputs (Zhou et al., 2024; Dong et al., 2024a). Our work complements existing deferral techniques by introducing a principled framework that optimizes this critical trade-off, enabling efficient processing of low-risk inputs while maintaining strong deferral guarantees where necessary.

## 6 CONCLUSION

In conclusion, while our proposed DDL framework demonstrates promising capabilities in balancing safety and efficiency through calibrated deferral thresholds, several important limitations warrant further investigation. The framework's current reliance on static distributional assumptions and the implicit relationship between model size and safety verification capabilities may not fully capture the complexities of real-world deployments. Additionally, deeper integration with complementary safety strategies and adaptation to black-box API scenarios remain important challenges to address. Nevertheless, our experimental results demonstrate DDL's significant contributions through its unique dual-size architecture and principled verification mechanism. The framework effectively enables dynamic resource allocation based on task complexity and safety requirements, achieving superior performance across multiple datasets and safety-critical scenarios. Detailed discussions of future work, limitations, impacts, and LLM usage are provided in Appendix Secs. J and K.

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

# A  LOSS FUNCTION DESIGN

Our proposed loss function $L(A_{\tau_1,\tau_2}(X), Y)$ is designed to provide theoretical guarantees while capturing three critical aspects of system performance. Given the bounded loss requirement for concentration inequalities and the need for Lipschitz continuity, we formulate the loss as a weighted combination:

$$L = \lambda_s \cdot S(X) + \lambda_e \cdot E(X, Y) - \lambda_t \cdot T(X) \tag{6}$$

where weights $\lambda_s$, $\lambda_e$, and $\lambda_t$ sum to unity to ensure boundedness in [0,1].

The safety assessment component $S(X)$ incorporates exponential penalties to enforce strict risk control:

$$S(X) = \sum_{p \in P} n_p \cdot e^{2e_p} \cdot w_p / N \tag{7}$$

Here, the processing paths $P = \{\text{large\_model}, \text{small\_model}, \text{human}\}$ represent our multi-stage routing system, with $n_p$ samples and error rate $e_p$ for each path $p$. The risk weights $w_p$ allow for path-specific penalty calibration, while $N$ normalizes the overall safety score.

For efficiency evaluation, we combine prediction accuracy with routing balance through:

$$E(X, Y) = (F_1 + D_s)/2 \tag{8}$$

where the F1-score measures prediction quality and the distribution score $D_s$ captures routing balance:

$$F_1 = \text{F1-score of predictions} \tag{9}$$
$$D_s = 1 - (|R_l - T_l| + |R_s - T_s|)/2 \tag{10}$$

The actual routing ratios $R_l$, $R_s$ and their targets $T_l$, $T_s$ ensure balanced workload distribution across models. To account for computational efficiency, we define the time cost metric:

$$T(X) = W_t + \sum_{i \in P} |R_i - T_i| \tag{11}$$

with weighted processing time:

$$W_t = R_l \cdot c_l + R_s \cdot c_s \cdot 0.5 + R_h \cdot e^{R_h} \tag{12}$$

The relative costs $c_l$ and $c_s$ are normalized by human review time, while the exponential term $e^{R_h}$ heavily penalizes excessive human review reliance. This formulation satisfies the theoretical requirements while incentivizing desired system behavior: the boundedness enables concentration inequalities for risk control, the continuous components ensure Lipschitz continuity. The thresholds $\tau_1$ and $\tau_2$ serve as calibration parameters to optimize this multi-objective trade-off between safety, efficiency, and computational cost.

# B  DETAILS OF THEORIES

## B.1  PROOF OF THEOREM 3.1

*Proof.* Under the null hypothesis $H_j : R(A_{\tau_1^{(j)}, \tau_2^{(j)}}) > \alpha$, the p-value $p_j$ is constructed such that it stochastically dominates a uniform distribution on the interval $[0, 1]$. This means that for any $u \in [0, 1]$, we have

$$P(p_j \leq u \mid H_j) \leq u.$$

In particular, setting $u = \delta/N$, it follows that

$$P(p_j \leq \delta/N \mid H_j) \leq \frac{\delta}{N}.$$

Since there are $N$ such hypotheses, we apply the union bound to ensure that the probability of falsely rejecting at least one true null hypothesis is at most

$$\sum_{j=1}^{N} P(p_j \leq \delta/N \mid H_j) \leq N \cdot \frac{\delta}{N} = \delta.$$

Therefore, with probability at least $1 - \delta$, none of the true null hypotheses are rejected when using a threshold of $\delta/N$.

Next, assume that there exists at least one pair $(\tau_1, \tau_2) \in \Lambda$ such that $R(A_{\tau_1, \tau_2}) \leq \alpha$. For this pair, the null hypothesis $H_j$ is false, meaning that the corresponding p-value $p_j$ does not necessarily stochastically dominate a uniform distribution and is expected to be smaller. By selecting the smallest p-value $p_{(1)}$, we are likely selecting a p-value corresponding to such a pair where $R(A_{\tau_1, \tau_2}) \leq \alpha$.

Since we only reject hypotheses with p-values below $\delta/N$, and we have at least one such valid pair, the probability that we correctly identify and reject the corresponding null hypothesis is at least $1 - \delta$. Consequently, the selected pair $(\hat{\tau}_1, \hat{\tau}_2)$ satisfies

$$R(A_{\hat{\tau}_1, \hat{\tau}_2}) \leq \alpha$$

with probability at least $1 - \delta$.

$\square$

This theorem assures that even a naive multiple testing approach (Teunissen, 2024), akin to Bonferroni correction (Weisstein, 2004), suffices to provide guaranteed risk with high probability given any safe pair $(\tau_1, \tau_2)$ selected based on the aforementioned p-value. The method only needs (1) boundedness of $L$ and (2) exchangeability between calibration and test samples, and does not depend on additional assumptions about the distribution of calibration and test data. Combined with the token embedding training that provides a reliable safety score, we thus have a method for configuring the two-stage decision pipeline to achieve finite-sample, distribution-free risk control at a pre-specified level $(\alpha, \delta)$, ensuring a safe and efficient large language model deployment.

In addition to Theorem 3.1, it is instructive to formalize several complementary results that highlight scenarios in which the two-stage threshold mechanism can be optimized or refined further, while still maintaining a finite-sample distribution-free guarantee. Below, we present three additional theorems with their motivations, statements, and proofs, incorporating more detailed calculations in each proof to illustrate the mechanics of the concentration bounds and structural assumptions. In practical situations, one of the thresholds, say $\tau_1$, may influence the risk in a monotonic or nearly monotonic fashion. For example, raising $\tau_1$ might reduce the false-positive rate of unsafe prompts at the small model but increase the number of escalations to the large model. If the true risk also decreases or remains bounded by increasing $\tau_1$, we can order threshold pairs accordingly and conduct a more efficient search in a single sequence rather than testing all pairs. This theorem provides a rigorous basis for such a monotonic search strategy, ensuring that once we identify a safe configuration at some level of $\tau_1$, all higher levels are also guaranteed safe.

### B.2    Proof of Theorem 3.2

**Assumption 4.** The loss function $L(A_{\tau_1, \tau_2}(x), y)$ is bounded, i.e.,

$$0 \leq L(A_{\tau_1, \tau_2}(x), y) \leq 1 \quad \text{for all } x, y, \tau_1, \tau_2.$$

This assumption holds for many common loss functions in classification. For example, the 0-1 loss used for accuracy naturally satisfies this bound as it only takes values in $\{0, 1\}$. Other examples include the calibrated surrogate losses like hinge loss after normalization $L(z, y) = \max(0, 1 - yz)/2$, or the sigmoid loss $L(z, y) = 1/(1 + \exp(yz))$. For fairness metrics, demographic parity difference and equal opportunity difference are also bounded in $[0, 1]$ after proper normalization.

**Assumption 5.** The risk function $R(\tau_1, \tau_2) = \mathbb{E}[L(A_{\tau_1, \tau_2}(X), Y)]$ is Lipschitz continuous with respect to the thresholds $\tau_1$ and $\tau_2$. Specifically, there exists a Lipschitz constant $K > 0$ such that for all $(\tau_1', \tau_2')$ and $(\tau_1'', \tau_2'')$ in $[0, 1]^2$,

$$|R(\tau_1', \tau_2') - R(\tau_1'', \tau_2'')| \leq K\Big(|\tau_1' - \tau_1''| + |\tau_2' - \tau_2''|\Big)$$

**Assumption 6.** The calibration samples $\{(X_i, Y_i)\}_{i=1}^n$ and test samples are exchangeable, i.e., they are independent and identically distributed (i.i.d.) according to the same distribution of $(X, Y)$. This standard assumption ensures that the performance on the calibration set generalizes to future test data.

**Lemma 1.** *Under Assumption 4 and Assumption 6, for a given grid $\Lambda_\Delta$ and confidence level $1 - \delta$, the following concentration inequality holds:*

$$P\left(\max_{(\tau_1, \tau_2) \in \Lambda_\Delta} \left|\hat{R}(\tau_1, \tau_2) - R(\tau_1, \tau_2)\right| \leq C\sqrt{\frac{\log|\Lambda_\Delta| + \log(1/\delta)}{n}}\right) \geq 1 - \delta,$$

*where $C > 0$ is a universal constant.*

*Proof of Lemma 1.* This lemma is established by applying the Hoeffding inequality to each grid point $(\tau_1, \tau_2) \in \Lambda_\Delta$. Since the loss function $L$ is bounded within $[0, 1]$ (Assumption 4) and the samples are i.i.d. (Assumption 6), Hoeffding's inequality can be directly applied. Specifically, for each $(\tau_1, \tau_2) \in \Lambda_\Delta$, we have:

$$P\Big(\left|\hat{R}(\tau_1, \tau_2) - R(\tau_1, \tau_2)\right| \geq \epsilon\Big)$$
$$\leq 2\exp\left(-2n\epsilon^2\right).$$

Applying the union bound over all $|\Lambda_\Delta|$ grid points, we obtain the following equation:

$$P\Big(\exists(\tau_1, \tau_2) \in \Lambda_\Delta : \left|\hat{R}(\tau_1, \tau_2) - R(\tau_1, \tau_2)\right| \geq \epsilon\Big) \leq 2|\Lambda_\Delta|\exp\left(-2n\epsilon^2\right).$$

Setting the right-hand side to be at most $\delta$, we solve for $\epsilon$ as follows:

$$2|\Lambda_\Delta|\exp\left(-2n\epsilon^2\right) \leq \delta \Rightarrow \epsilon \geq \sqrt{\frac{\log(2|\Lambda_\Delta|/\delta)}{2n}}.$$

Choosing $C$ appropriately (e.g., $C \geq \sqrt{1}$ to account for constants in inequalities), we simplify the bound to the following equation:

$$\epsilon = C\sqrt{\frac{\log|\Lambda_\Delta| + \log(1/\delta)}{n}}.$$

Thus, with probability at least $1 - \delta$, the maximum deviation across all grid points is bounded as stated. $\square$

*Proof of Theorem 3.2.* We begin by applying Lemma 1 to obtain a high-probability bound on the maximum deviation between the empirical risk $\hat{R}(\tau_1, \tau_2)$ and the true risk $R(\tau_1, \tau_2)$ over the grid $\Lambda_\Delta$. Specifically, with probability at least $1 - \delta$,:

$$\max_{(\tau_1, \tau_2) \in \Lambda_\Delta} \left|\hat{R}(\tau_1, \tau_2) - R(\tau_1, \tau_2)\right| \leq C\sqrt{\frac{\log|\Lambda_\Delta| + \log(1/\delta)}{n}}.$$

This event is referred to as the high-probability event. Next, consider any threshold pair $(\tau_1, \tau_2) \in [0, 1]^2$. By the construction of the grid $\Lambda_\Delta$, there exists a corresponding grid point $(\bar{\tau}_1, \bar{\tau}_2) \in \Lambda_\Delta$:

$$|\tau_1 - \bar{\tau}_1| \leq \Delta \quad \text{and} \quad |\tau_2 - \bar{\tau}_2| \leq \Delta.$$

Using the Lipschitz continuity of the risk function $R(\tau_1, \tau_2)$ (Assumption 5), we can bound the difference in risk between $(\tau_1, \tau_2)$ and $(\bar{\tau}_1, \bar{\tau}_2)$:

$$R(\tau_1, \tau_2) \leq R(\bar{\tau}_1, \bar{\tau}_2) + K\left(|\tau_1 - \bar{\tau}_1| + |\tau_2 - \bar{\tau}_2|\right) \leq R(\bar{\tau}_1, \bar{\tau}_2) + 2K\Delta.$$

On the high-probability event established by Lemma 1, we further bound $R(\bar{\tau}_1, \bar{\tau}_2)$ using the empirical risk:

$$R(\bar{\tau}_1, \bar{\tau}_2) \leq \hat{R}(\bar{\tau}_1, \bar{\tau}_2) + C\sqrt{\frac{\log|\Lambda_\Delta| + \log(1/\delta)}{n}}.$$

Combining these inequalities, we obtain:

$$R(\tau_1, \tau_2) \leq \hat{R}(\bar{\tau}_1, \bar{\tau}_2) + C\sqrt{\frac{\log|\Lambda_\Delta| + \log(1/\delta)}{n}} + 2K\Delta.$$

Since $(\bar{\tau}_1, \bar{\tau}_2) \in \Lambda_\Delta$, the above inequality implies that for any $(\tau_1, \tau_2) \in [0, 1]^2$, the true risk is bounded by the empirical risk at the nearest grid point plus the deviation terms.

Now, if there exists a grid point $(\tilde{\tau}_1, \tilde{\tau}_2) \in \Lambda_\Delta$ satisfying:

$$\hat{R}(\tilde{\tau}_1, \tilde{\tau}_2) + C\sqrt{\frac{\log|\Lambda_\Delta| + \log(1/\delta)}{n}} + 2K\Delta \leq \alpha,$$

then, for any $(\tau_1, \tau_2) \in [0, 1]^2$ close to $(\tilde{\tau}_1, \tilde{\tau}_2)$ within $\Delta$, we have:

$$R(\tau_1, \tau_2) \leq \hat{R}(\tilde{\tau}_1, \tilde{\tau}_2) + C\sqrt{\frac{\log|\Lambda_\Delta| + \log(1/\delta)}{n}} + 2K\Delta \leq \alpha.$$

Therefore, with probability at least $1 - \delta$, any threshold pair within $\Delta$ distance of a qualifying grid point will satisfy the risk constraint $R(\tau_1, \tau_2) \leq \alpha$. □

### B.3 THEOREM OF MONOTONICITY PROPERTIES

**Theorem B.1.** *Suppose we have a list of candidate pairs*

$$\left\{\left(\tau_1^{(1)}, \tau_2^{(1)}\right), \left(\tau_1^{(2)}, \tau_2^{(2)}\right), \ldots, \left(\tau_1^{(N)}, \tau_2^{(N)}\right)\right\}$$

*ordered so that $\tau_1^{(1)} \leq \tau_1^{(2)} \leq \cdots \leq \tau_1^{(N)}$. Assume for each $k = 1, \ldots, N-1$,*

$$\tau_1^{(k)} \leq \tau_1^{(k+1)} \implies$$
$$R\left(A_{\tau_1^{(k)}, \tau_2^{(k)}}\right) \geq R\left(A_{\tau_1^{(k+1)}, \tau_2^{(k+1)}}\right).$$

*Then, if $R\left(A_{\tau_1^{(k)}, \tau_2^{(k)}}\right) \leq \alpha$ holds for some $k$, all pairs $\left(\tau_1^{(j)}, \tau_2^{(j)}\right)$ with $j \geq k$ also satisfy $R\left(A_{\tau_1^{(j)}, \tau_2^{(j)}}\right) \leq \alpha$. Consequently, a sequential testing procedure that begins with $k = 1$ and proceeds upward can terminate as soon as it encounters a safe pair, thereby reducing the multiplicity penalty and improving power.*

*Proof.* First, the condition implies that whenever $\tau_1$ is increased from $\tau_1^{(k)}$ to $\tau_1^{(k+1)}$, the risk cannot increase; more precisely, $R(A_{\tau_1^{(k+1)}, \tau_2^{(k+1)}}) \leq R(A_{\tau_1^{(k)}, \tau_2^{(k)}})$. Thus, if for some index $k$, $R(A_{\tau_1^{(k)}, \tau_2^{(k)}}) \leq \alpha$, it follows that

$$R\left(A_{\tau_1^{(k+1)}, \tau_2^{(k+1)}}\right) \leq R\left(A_{\tau_1^{(k)}, \tau_2^{(k)}}\right) \leq \alpha, \tag{13}$$

and inductively, every pair with a higher index is also safe. This enables a fixed-sequence test that checks hypotheses $H_1, H_2, \ldots, H_N$ in order, where $H_k$ denotes $R(A_{\tau_1^{(k)}, \tau_2^{(k)}}) > \alpha$. Once $H_k$ is rejected, all $H_{k'}$ with $k' > k$ are automatically false nulls (meaning the risk is at most $\alpha$) under the given monotonicity assumption, so no further testing is required. By not testing the latter hypotheses, we reduce the chance of incurring false rejections and preserve more statistical power. □

Table 4: Dataset Statistics

| Dataset Category | Sample Size | Avg. Token Length | Distribution |
|---|---|---|---|
| **Token-based Classifier Dataset (Total: 30,000)** | | | |
| Safe Samples | 15,000 | – | MMLU (5,000) BoolQ (5,000) WikiQA (5,000) |
| Unsafe Samples | 15,000 | – | Modified samples with injected unsafe content |
| **Evaluation Dataset (Total: 7,500)** | | | |
| MMLU-derived | 2,500 | 67.3 (Safe) 98.4 (Unsafe) | Medicine (25%) Law (20%) Engineering (15%) Humanities (25%) Social Sciences (15%) |
| BoolQ-derived | 2,500 | 42.8 (Safe) 88.6 (Unsafe) | News (45%) Wikipedia (35%) Web Content (20%) |
| WikiQA-derived | 2,500 | 45.6 (Safe) 92.8 (Unsafe) | History (30%) Science (25%) Current Events (20%) Culture (25%) |

## B.4 THEOREM OF DATA-DRIVEN THRESHOLD SELECTION

**Theorem B.2.** *Let $(\tau_1^{(1)}, \tau_2^{(1)}), \ldots, (\tau_1^{(N)}, \tau_2^{(N)})$ be candidate threshold pairs, and let $H_j$ : $R(\tau_1^{(j)}, \tau_2^{(j)}) > \alpha$ denote the null. Suppose each $H_j$ is tested at level $\delta/N$ by a valid p-value $p_j$. Let $\mathcal{R}$ be the set of all indices $j$ for which $H_j$ is rejected. With probability at least $1 - \delta$, no true null is rejected, so every $j \in \mathcal{R}$ satisfies $R(\tau_1^{(j)}, \tau_2^{(j)}) \leq \alpha$. Hence, we may define*

$$\hat{j} := \arg\min_{j \in \mathcal{R}} f\left(\tau_1^{(j)}, \tau_2^{(j)}\right), \tag{14}$$

*where $f$ is an arbitrary secondary objective (e.g., cost, accuracy on normal queries), and conclude that $R(\tau_1^{(\hat{j})}, \tau_2^{(\hat{j})}) \leq \alpha$.*

*Proof.* Define a familywise testing procedure with overall level $\delta$ by assigning $\delta/N$ to each $j$. Because each $p_j$ stochastically dominates $\mathrm{Uniform}(0, 1)$ under $H_j$, the expected number of false rejections is bounded, and by the union bound or a standard FWER control argument, the probability of rejecting any true null is at most $\delta$. On this event that no true null is rejected, every rejected index $j \in \mathcal{R}$ must be a false null, i.e., $R(\tau_1^{(j)}, \tau_2^{(j)}) \leq \alpha$. Consequently, picking an index $\hat{j} \in \mathcal{R}$ to minimize or maximize any function $f(\tau_1, \tau_2)$ does not affect the validity of the risk constraint, so $R(\tau_1^{(\hat{j})}, \tau_2^{(\hat{j})}) \leq \alpha$ with probability at least $1 - \delta$. No additional correction is needed because the data-driven selection in equation 14 occurs purely within the safe set $\mathcal{R}$. $\square$

Theorem B.2 confirms that data-driven selection among all threshold pairs passing the safety test preserves the distribution-free risk guarantee, thereby allowing flexibility in optimizing other performance metrics while guaranteeing $R(A_{\hat{\tau}_1, \hat{\tau}_2}) \leq \alpha$. The synergy between these four results—Theorem 3.1, Theorems B.1, 3.2, and B.2—provides a rigorous methodology for configuring the two-stage pipeline to detect and block malicious queries with finite-sample confidence in safety level. This closes the theoretical loop between token-embedding-based training and threshold calibration in a distribution-free manner.

## C  DETAILS OF TOKEN CLASSIFICATION DATASET

To create a comprehensive training dataset for safety classification, we combine multiple high-quality data sources through a carefully designed process. Each example is augmented with the same special safety classification token $t$ and annotated with binary labels. The core safe queries are sampled from three widely used question-answering datasets: MMLU (Hendrycks et al., 2020), BoolQ (Clark et al., 2019), and WikiQA (Yang et al., 2015). These datasets provide naturally occurring, well-formed questions across diverse topics and domains. For the unsafe portion, we leverage the BeaverTails dataset, which contains a curated collection of potentially harmful queries spanning various safety concerns.

The training set construction follows a systematic approach to ensure quality and balance. We first randomly sample 15,000 questions from the safe sources, maintaining equal proportions from MMLU, BoolQ, and WikiQA, with approximately 5,000 samples from each dataset. This balanced sampling ensures diverse coverage across different question types, including factual queries in WikiQA, complex reasoning questions in MMLU, and yes/no questions in BoolQ.

To generate the unsafe samples, we employ a novel hybrid approach - we randomly select unsafe prompts from BeaverTails and carefully inject them into regular questions at random positions. This injection method preserves the semantic structure of the original questions while introducing safety-critical elements, creating realistic examples of potentially harmful queries. Each unsafe prompt is strategically inserted with surrounding context markers to maintain natural language flow. For example, a regular question about technology might be augmented with potentially harmful instructions, or a general knowledge query might be modified to include privacy-violating elements.

The injection process is carefully controlled to ensure the resulting queries remain coherent and representative of real-world scenarios. We utilize random position insertion to create varied examples where unsafe content appears at different locations within the query - beginning, middle, or end. This variation helps train models to identify safety concerns regardless of their position in the query. Additionally, we maintain the original semantic structure of the base questions to ensure the unsafe elements are naturally integrated rather than appearing as obvious modifications.

As shown in Table 4, the final training set consists of 30,000 samples, evenly split between safe and unsafe categories (15,000 each). This substantial dataset size enables robust model training while maintaining balanced representation across different safety categories and question types. The large scale of the dataset also enables better coverage of edge cases and subtle variations in how unsafe content may manifest in real-world queries. Each sample is annotated with binary safety labels (0 for unsafe, 1 for safe) and includes the full query text, enabling straightforward use in classification tasks.

## D  EVALUATION DATASET STATISTICS

As shown in Table 4, the evaluation set encompasses 7,500 meticulously curated samples from three major datasets. We carefully maintain balanced distributions and quality metrics across each source to ensure comprehensive coverage of different question types and safety scenarios.

**MMLU-derived Samples (2,500)**  The MMLU portion contributes 2,500 samples (33.3% of total), evenly split between safe and unsafe variants. The safe samples (1,250) are drawn directly from the MMLU test split, covering professional and academic domains including medicine (25%), law (20%), engineering (15%), humanities (25%), and social sciences (15%). These questions average 67.3 tokens in length and feature domain-specific terminology. The unsafe variants (1,250) are generated through controlled injection of safety-critical content, resulting in modified questions averaging 98.4 tokens. The safety modifications maintain the original domain context while introducing carefully calibrated unsafe elements across our taxonomy of safety concerns.

**BoolQ-derived Samples (2,500)**  The BoolQ validation set provides another 2,500 samples (33.3%), with equal safe and unsafe distributions. The safe portion (1,250 samples) consists of natural yes/no questions drawn from real user queries, with domain coverage spanning news articles (45%), Wikipedia passages (35%), and general web content (20%). These questions maintain an average length of 42.8 tokens and preserve the original boolean question structure. The corresponding unsafe samples (1,250) are created through our injection protocol, resulting in modified questions averaging

88.6 tokens. While maintaining the yes/no answerable format, these variants incorporate safety concerns that test the model's ability to identify potentially harmful content within natural question structures.

**WikiQA-derived Samples (2,500)** The WikiQA test split contributes the final 2,500 samples (33.3%), again balanced between safe and unsafe cases. The safe samples (1,250) focus on factual questions distributed across history (30%), science (25%), current events (20%), and culture (25%), with an average length of 45.6 tokens. These questions are characterized by their focus on specific factual information and encyclopedia-style content. The unsafe variants (1,250) are generated through careful modification of these base questions, averaging 92.8 tokens in length. These modifications preserve the factual querying style while introducing safety-critical elements that test the model's ability to identify harmful content in knowledge-seeking contexts.

## E    DETAILS ABOUT METRICS

For safety, we compute a risk score that combines the weighted false positive rate (FPR) and false negative rate (FNR), where false positives represent unsafe content being incorrectly classified as safe. For accuracy, we measure both the overall classification performance using F1-score and model-specific accuracies for different routing paths. To evaluate efficiency, we use a time cost metric $T = \sum_{i \in \{L,S,H\}} r_i \cdot t_i$, where $r_i$ is the routing ratio and $t_i$ is the average processing time for each path (large model, small model, or human review), normalized by the baseline cost of full human review to obtain a relative efficiency score. Error Rate ($E_r$) is defined as the percentage of malformed or harmful responses in test samples: $E_r = \frac{N_{error}}{N_{total}} \times 100\%$, where $N_{error}$ is the number of responses that are unsafe, illegal or do not conform to the expected format, and $N_{total}$ is the total number of test samples. A lower error rate indicates better model reliability and output consistency, which is crucial for real-world applications where safety constraints are mandatory.

## F    MORE DETAILS ABOUT IMPLEMENTATION

All experiments are conducted on 4 NVIDIA Tesla V100 GPUs (32GB memory each). For reproducibility, we run each experiment 10 times with fixed random seeds and report the mean values. We implement our framework using PyTorch and adopt efficient batched processing with batch size $b = 256$. For the trainable token-based safety classifier, we utilize a base encoder with 24 transformer layers, where the special token embedding dimension is set to $d = 1536$. The model architecture consists of three fully-connected layers ($2048 \rightarrow 1024 \rightarrow 512$) with GELU activation and dropout rate $p_d = 0.1$ for regularization. Training proceeds using Adam optimizer ($\beta_1 = 0.9$, $\beta_2 = 0.999$) with learning rate $\eta = 1e^{-3}$, cosine decay scheduling, and weight decay $\lambda_w = 0.01$. During training, we inject Gaussian noise with learnable variance parameter $\sigma$ to improve robustness. The model outputs calibrated probability scores through a sigmoid layer. We incorporate early stopping with patience = 5 epochs and maintain the exponential moving average of model parameters with decay rate as 0.999. The training process typically converges within 100 epochs on our 230K sample dataset. The calibration set consists of 1K randomly selected examples.

For the dual-stage deferral mechanism, the system handles batched processing with maximum batch size $b_i = 32$, implementing efficient token-based computation. We employ an asynchronous queue-based architecture for routing between models with configurable timeout $t_{\max} = 30s$ and exponential backoff retry logic (max 3 retries).

The calibration module employs our comprehensive loss function $\mathcal{L}$ defined as:

$$\mathcal{L} = \lambda_s \cdot S(X) + \lambda_e \cdot E(X,Y) - \lambda_t \cdot T(X) \tag{15}$$

where $\lambda_s = 0.4$, $\lambda_e = 0.3$, and $\lambda_t = 0.3$ are weights for safety $S(X)$, efficiency $E(X,Y)$, and time cost $T(X)$ respectively. Here, $S(X) = \sum_{p \in P} n_p \cdot e_p \cdot w_p / N$ evaluates safety with $n_p$ samples in path $p$, error rate $e_p$, and risk weight $w_p$. Importantly, the efficiency term $E(X,Y)$ is computed directly from the F1-score, which captures task accuracy. Thus, accuracy is an intrinsic part of the optimization objective, alongside safety and efficiency. By adjusting the weight $\lambda_e$, practitioners can

emphasize accuracy more strongly in tasks with stricter performance requirements, allowing flexible trade-offs among the three objectives.

The optimization is constrained by false positive rate and false negative rate thresholds:

$$FPR \leq \alpha \quad FNR \leq \delta \tag{16}$$

In our implementation, we set $\alpha = 0.05$ and $\delta = 0.05$ in Sec 2.4 as the constraints for false positive and negative rates. We implement exhaustive grid search over 6,000 uniformly spaced threshold candidates in $[0.1, 0.9]$ ($\Delta = 0.01$) with statistical guarantees for maintaining these constraints.

In our implementation, we set $\alpha = 0.05$ and $\delta = 0.05$ in Sec 2.4 as the constraints for false positive and negative rates. We implement exhaustive grid search over 6,000 uniformly spaced threshold candidates in $[0.1, 0.9]$ ($\Delta = 0.01$) with statistical guarantees for maintaining these constraints.

Our risk-controlled routing system maintains detailed statistics, including per-model latencies, routing distributions, and error rates by confidence score buckets. The system features configurable risk constraints and can adapt thresholds based on observed error patterns through sliding window averages (window size $w = 1000$). We validate performance using 10-fold cross validation and employ parameter-efficient fine-tuning procedures with learning rate $\eta_f = 5e^{-5}$ for domain adaptation. The complete pipeline maintains comprehensive logging of all routing decisions and predictions for post-hoc analysis. The system demonstrates robust performance across different input distributions while maintaining the desired risk control guarantees specified in our theoretical framework.

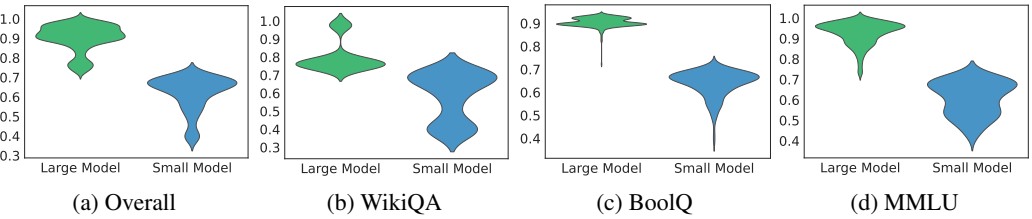

|  (a) Overall | (b) WikiQA | (c) BoolQ | (d) MMLU |

Figure 4: Estimated risk distribution of queries processed by different model sizes across datasets. Our DDL mechanism evaluates query risk levels and routes them to appropriate models.

## G RISK DISTRIBUTION AND ROUTING ANALYSIS

As illustrated in Figure 4, taking DDL (Qwen 1.5B–7B) as an example, we use the output confidence scores from the token classifier trained on Qwen-72B as the estimated risk scores for each query, serving as the estimated ground truth. Our experimental results reveal an intriguing pattern in risk assessment and task routing. For simple queries (typically assigned higher risk scores), Qwen-7B naturally handles more tasks with a high estimated risk score. This pattern is particularly evident in BoolQ (Figure 4c) and MMLU(Figure 4d), where straightforward yes/no questions are often assigned higher risk scores and routed to Qwen-7B. Conversely, for more complex or ambiguous queries (typically assigned lower risk scores), the system tends to route tasks to seek verification of powerful DeepSeek. This counterintuitive yet effective routing strategy is especially apparent in WikiQA (Figure 4b), where complex domain knowledge questions are often assigned lower risk scores, leading to more conservative handling through verification of DeepSeek. The experimental results also show distinct routing distributions across different tasks. For BoolQ's straightforward questions, our system routed 68.2% of queries to Qwen-7B, 25.3% to Qwen-1B, and only 6.5% required DeepSeek verification. In MMLU, which contains more complex domain knowledge questions, 56.7% were processed by Qwen-7B, while a larger portion (34.9%) required DeepSeek verification, and 8.4% were handled by Qwen-1B. Most notably, in WikiQA with its open-ended queries, DeepSeek verification was required for 50.3% of cases, while Qwen-7B handled 37.8% and Qwen-1B processed 11.9%. These distribution patterns demonstrate how query complexity naturally influences the risk assessment and subsequent routing outcomes, validating the effectiveness of our dual-stage deferral mechanism in balancing between computational efficiency and safety guarantees.

## H   EVALUATIONS ON HARMBENCH

To evaluate DDL's robustness against adversarial attacks, we conduct additional evaluations based on HarmBench (Mazeika et al., 2024), a standardized evaluation framework for automated red teaming of Large Language Models that enables systematic assessment of attack and defense methods while introducing rigorous criteria previously missing in the field.

We design two settings: (i) Scenario A (baseline) uses our original test set, and (ii) Scenario B mixes 500 attack prompts with 1,500 benign queries (25% attack ratio). As a supplementary check, we also test with a deployment-oriented

Table 5: DDL performance comparison between baseline and HarmBench adversarial scenarios.

| Metrics | Scenario A (Baseline) | Scenario B (HarmBench) | Difference |
|---|---|---|---|
| *Safety Metrics* | | | |
| False Negative Rate (%) | 4.2 | 4.5 | +0.3 |
| False Positive Rate (%) | 15.4 | 16.6 | +1.2 |
| *Efficiency Metrics* | | | |
| Avg Response Time (ms) | 42 | 47 | +5 |
| M1 Processing (%) | 55 | 48 | -7 |
| M2 Processing (%) | 30 | 35 | +5 |
| M3 (Human) Review (%) | 15 | 17 | +2 |

10:1 ratio (100 attacks, 1,000 benign). As shown in Table 5, DDL achieves a 4.5% FNR in Scenario B, only a 0.3% increase over baseline. The higher FPR mainly stems from Stage 1 (M1), the lightweight classifier: benign queries mis-scored as risky are automatically escalated to M2 or M3, preventing safety failures. Under attack pressure, routing adapts by reducing M1 share (55% to 48%) and raising M2 share (30% to 35%), showing dynamic adjustment to adversarial complexity.

While FPR rises (up to 16.6% in the supplementary 10:1 test), this reflects our conservative thresholds for keeping FNR below 5%. We view this as an acceptable cost: in high-stakes domains, avoiding missed attacks outweighs extra false alarms, which mainly add modest latency. Moreover, the parameters $\alpha$ and $\delta$ make this trade-off explicitly tunable for different risk tolerances. Overall, HarmBench results show that DDL maintains strong safety under stronger adversarial scenarios, with transparent and controllable efficiency–safety trade-offs suitable for real-world deployment.

## I   EFFICIENCY ANALYSIS

We conduct a comprehensive efficiency analysis to validate the computational benefits of our approach across both classifier architecture choices and calibration overhead.

**Classifier Architecture Comparison.** To justify our lightweight classifier design, we compare three different architectures for the first-stage assessment task. The results demonstrate that more complex architectures provide minimal accuracy gains while computational costs increase dramatically:

Table 6: Comparison of different classifier architectures for first-stage assessment.

| Classifier Method | Accuracy | Inference Latency |
|---|---|---|
| DDL-Linear (Ours) | 81.5% | ∼0.2 ms |
| DDL-MLP (2-layer) | 82.1% | ∼0.5 ms |
| DDL-CLS (using [CLS] token) | 82.8% | ∼3.5 ms |

The DDL-MLP and DDL-CLS approaches achieve only 0.6% and 1.3% accuracy improvements respectively, while incurring 2.5× and 17.5× increases in latency. The DDL-CLS approach is particularly costly as it requires a full forward pass to extract the [CLS] token representation. This validates our design choice: for a "fast routing" task, an extremely lightweight classifier achieves the optimal efficiency-performance trade-off.

**Calibration Overhead Analysis.** The threshold calibration process is a one-time, offline procedure that incurs no runtime overhead. In our experiments using 1000 calibration samples on a single NVIDIA V100 GPU, the entire calibration process takes approximately 30 minutes. This one-time cost is negligible for a system intended for long-term online service, as the calibrated thresholds can be reused across millions of inference requests without requiring recalibration.

## J    IMPACTS, LIMITATIONS, AND FUTURE WORK

In this section, we discuss the broader impacts of our work, acknowledge its limitations, and outline promising directions for future research.

The DDL framework offers significant broader impacts beyond its technical contributions. By optimizing the trade-off between safety and efficiency, it can lower the computational barrier for deploying safe AI systems, making advanced LLM technology more accessible to organizations with limited resources. In critical domains such as education and healthcare, the dual-stage risk control mechanism can enhance system reliability and user trust. However, we acknowledge that no automated system is infallible. The framework may still produce incorrect judgments in edge cases. Therefore, we advocate for a tiered deployment strategy, especially in high-stakes scenarios, which includes maintaining meaningful human oversight and implementing continuous monitoring and feedback loops to ensure robust and responsible operation.

Despite the promising results of our current DDL framework, several important limitations remain. The dual-stage deferral mechanism relies on the assumption that larger models consistently provide stronger safety verification. In practice, this relationship can be more nuanced and may depend on factors like training data quality or model architecture, suggesting that a carefully optimized smaller model could also be effective in certain contexts. Our threshold calibration procedure similarly adopts static distributional assumptions, which may not fully capture dynamically evolving query patterns and shifting risk landscapes in real-world deployments. This highlights the need for more adaptive approaches. In addition, the current token-based classifier, while efficient, operates within a vertical pipeline. This design lacks deeper integration with complementary horizontal safety strategies, such as adversarial training, prompt optimization, or human-AI collaboration, each of which could enhance robustness in different ways. Finally, deployment with black-box APIs, where internal model signals are unavailable, poses a particular challenge, limiting the direct application of our current method.

The limitations identified above naturally lead to several exciting directions for future research. A key priority is developing more sophisticated criteria for model selection in the verification pipeline that go beyond simple parameter counts, potentially incorporating models with specialized safety capabilities or unique architectural components. Another important direction is exploring how to effectively combine our vertical deferral mechanism with horizontal safety strategies (Ma et al., 2025). For instance, this could involve integrating adversarial defense techniques (Kumar, 2024) to enhance robustness, incorporating safety-aware prompt engineering (Kumar et al., 2023) to guide model outputs, or developing hybrid human-AI verification protocols (Ibrahim et al., 2024) for high-stakes scenarios. Furthermore, addressing the challenge of distributional shifts requires developing adaptive calibration methods. If the online prompt distribution changes significantly, re-calibration with new representative samples would be needed to maintain risk control, which in practice can be implemented through periodic monitoring and automated pipelines. We believe such a holistic approach, bridging vertical and horizontal safety mechanisms, could lead to more comprehensive and reliable safety systems for large language model deployment.

## K    LLM USAGE DISCLOSURE

In accordance with ICLR 2026 policy, we disclose our use of LLMs in this work. We employed Claude (Anthropic) to aid in polishing the writing of this manuscript, specifically for improving clarity, grammar, and sentence structure across various sections, though all technical content, experimental results, and scientific contributions remain entirely our own work. During the research process, we also utilized LLMs for brainstorming and refining research ideas, particularly in generating diverse query examples (Section 4.1 and Appendix C), systematically transforming benign queries into safety-critical versions for evaluation datasets, and exploring different safety categories and attack scenarios. Additionally, we employed LLMs to assist in checking and refactoring our implementation code, including reviewing the dual-stage deferral mechanism implementation for potential optimizations, suggesting improvements to the threshold calibration procedures, and identifying potential edge cases in our safety classifier. All LLM-generated content was carefully reviewed, validated, and modified by the authors, and the core algorithmic contributions, theoretical analysis, experimental design, and empirical evaluations were conducted independently by the research team without LLM assistance.

